

# SFedChain: blockchain-based federated learning scheme for secure data sharing in distributed energy storage networks

Mingming Meng and Yuancheng Li

School of Control and Computer Engineering, North China Electric Power University, Beijing, China

## ABSTRACT

The intelligence of energy storage devices has led to a sharp increase in the amount of detection data generated. Data sharing among distributed energy storage networks can realize collaborative control and comprehensive analysis, which effectively improves the clustering and intelligence. However, data security problems have become the main obstacle for energy storage devices to share data for joint modeling and analysis. The security issues caused by information leakage far outweigh property losses. In this article, we first proposed a blockchain-based machine learning scheme for secure data sharing in distributed energy storage networks. Then, we formulated the data sharing problem into a machine-learning problem by incorporating secure federated learning. Innovative verification methods and consensus mechanisms were used to encourage participants to act honestly, and to use well-designed incentive mechanisms to ensure the sustainable and stable operation of the system. We implemented the scheme of SFedChain and experimented on real datasets with different settings. The numerical results show that SFedChain is promising.

# INTRODUCTION

With the rise of a new round of energy revolution, energy and information are highly integrated. Energy storage as one of the areas with the most large-scale development potential in renewable energy, generates massive amounts of data with the improvement of informatization and intelligence. While data derives value, its information security issues have also received extensive attention (*Stoyanova et al., 2020*; *Cook et al., 2017*). Data leakage problems may occur in terminals, networks, storage, and the cloud, *etc*, which will cause serious obstacles to the construction of power information network security. In this regard, the traditional privacy protection strategy can be mainly divided into the privacy protection of input data and output data. The privacy protection of input data is mainly based on publishing anonymous data, such as k-anonymity, l-diversity, t-closeness, and differential privacy. k-anonymity, l-diversity and t-closeness (*Brickell & Shmatikov, 2008*; *Sei et al., 2019*). They usually replace the sensitive information contained in the data with randomly generated data or remove it directly. However, if the attacker has sufficient background knowledge of the original data, or attacks through inference or other methods,

Corresponding author
Yuancheng Li, ncepua@163.com

it will not be able to effectively protect the confidential information. Differential privacy (*Dwork, 2008*) technology achieves a balance between model performance and privacy protection by adding noise to the model or generated results, it is considered a reliable privacy protection method. *Yin et al. (2018)* applied differential privacy technology to hide the original trajectory and location data of the information by adding noise to the selected data in the location information tree model, which protects the location privacy of big data in the sensor network. The experimental results of *Hitaj, Ateniese & Pérez-Cruz (2017)* show that a data privacy protection strategy that only incorporates differential privacy may leak original data using generative adversarial network (GAN) learning. In order to prevent the inference of the original confidential data through the intermediate state information during the training of the Latent Dirichlet Allocation model, *Zhao et al. (2021)* proposed a privacy protection algorithm, Hierarchical Dirichlet Process-Latent Dirichlet Allocation (HDP-LDA) based on differential privacy, however, the algorithm is only applicable to a single model scenario and is not universal, so it is difficult to be effectively promoted. Therefore, how to improve the availability of data under the premise of protecting data privacy remains to be further studied.

The privacy protection of output data is mainly to pertube or audit the result (*Aggarwal, 2005*) such as association rule hiding, query auditing and classification accuracy. The existing association rule hiding technology (*Gkoulalas-Divanis & Verykios, 2009*; *Wu & Wang, 2008*) directly operates on the original transaction dataset. When the transaction dataset is relatively large, the time utilization rate will be relatively low, at the same time, it is difficult to achieve a good compromise between sensitive information hiding and data quality by artificially adding rules to the original transaction dataset to hide sensitive information. In *Hou et al. (2018)* and *Thomas (2007)*, the authors provide effective query audit algorithms and frameworks that leverage security review mechanisms for system privacy protection and access control. The classification accuracy improvement method (*Samanthula, Elmehdwi & Jiang, 2015*) achieves privacy protection by deforming confidential data when the classification accuracy of confidential data is close to that of reconstructed data, but the existence of a large amount of heterogeneous data and the limitations of restrictions make this method difficult to achieve a wide range of applications.

Data privacy protection algorithm based on deep learning can further improve data availability and prevent the risk of data leakage more efficiently compared with traditional data privacy protection strategies. Therefore, many excellent models for deep learning privacy protection have been proposed. Federal Learning (*Bonawitz et al., 2019*) stands out for its unique privacy policy. Multiple collaborators do not need to upload their raw data to the central server for iterative training during the deep learning model training process and get better training results than their respective local models. *Konečný et al. (2016)* proposed an efficient optimization algorithm to deal with the statistical heterogeneous problem of data in federated learning. *Fallah, Mokhtari & Ozdaglar (2020)* proposed personalized federated learning, which can be better done by localizing the global model by using a local data structure. Nevertheless, traditional federal learning techniques still have some privacy leakage problems due to the existence of curious parameter server or dishonest participants. *Nasr, Shokri & Houmansadr (2018)* indicates that secret membership information can be

obtained by performing member inferring attacks. *Zhu & Han (2020)* uses the depth gradient leakage algorithm to reduce the difference between the virtual gradient and the real gradient to obtain private data.

Due to the decentralization of energy storage devices and the confidentiality of generated data, how to protect privacy while collecting data is a key issue. Recently, the issue of multi-party data sharing has received widespread attention. For data sharing on distributed data streams, *Dong et al. (2015)* proposed a scheme for safe sharing of sensitive data on big data platforms. *Huang et al. (2021)* designed an accountable and efficient data sharing scheme ADS for industrial IoT, which can punish participants with data leakage problems. It is worth noting that Blockchain (*Huh, Cho & Kim, 2017*), as a decentralized, tamper-proof, and traceable distributed ledger technology, effectively guarantees the confidentiality of data and the security of data sharing by using consensus protocols. Due to data trust and security issues in the edge computing environment, *Ma et al. (2020b)* proposed a blockchain-based edge computing trusted data management scheme BlockTDM. Based on blockchain technology, *Ma et al. (2020a)* realized the secure utilization and decentralized management of big data in the Internet of Things. In the above work, a consensus algorithm that achieves the consistency of all participating nodes is indispensable as a key technology. In *Zheng et al. (2017)*, miners need to solve tedious mathematical problems and compete to produce blocks, which seriously affects the efficiency of the system, so it is not suitable for scenarios with frequent transactions.

Despite extensive research has been conducted on distributed multi-party data sharing, there are still two serious problems that have received less attention so far. The first is that the existing work usually targets the attack threats of the central server or collaborators, while ignoring the model quality problems caused by dishonest collaborators destroying the joint modeling process. The second is that participants' concerns about data privacy leakage in the process of distributed multi-party data sharing have led to the continuous decline of users' willingness to share data.

To the end, there are many challenges in distributed multi-party collaborative data sharing in distributed energy storage networks. We have established a new mechanism to ensure secure data sharing between collaborators who do not trust each other, and proposed a scheme based on blockchain and federated learning named SFedChain. Privacy protection and data sharing are carried out in the joint modeling by encrypting the original information, which can ensure the confidentiality of collaborators' data, the traceability of shared events, and the robustness of the training model. Specifically, we adopt the "Three Chains in One" approach to ensure the secure storage of data, auditability, and traceability. In addition, the use of encryption technology provides a further guarantee for the secure sharing of parameters. The adoption of novel consensus algorithms and incentive mechanisms and the use of election collaborators for parameter aggregation can effectively improve the security of the system and maximize the benefits of the system. To sum up, The specific contributions of this paper are as follows:

1. We proposed SFedChain, a novel distributed multi-party data sharing collaboration training scheme, which effectively reduces the risk of data leakage and achieves secure data sharing in the process of joint modeling.

2. SFedChain not only protects the privacy of data holders, but also realizes the secure storage of data, and the auditability and traceability of the sharing process. The adoption of efficient consensus algorithms and incentive mechanisms promotes collaborators to act honestly in the joint modeling process, thereby generating a high-performance joint modeling model.

3. We implemented SFedChain prototype and evaluate its performance in terms of training accuracy and training time. We also evaluate the effectiveness of our proposed model with benchmark, open real-world datasets for data categorization.

The rest of the article is organized as follows: in 'System Model', we present our system model. In 'Construction of Sfedchain Scheme', we give implementation details of SFedChain. In 'Security Analysis and Performance Evaluation', we present security analysis for our proposed scheme, and evaluate the performance of the SFedChain. Finally, 'Conclusion' summarizes this article.

## SYSTEM MODEL

In this article, we assume a joint modeling scenario involving multiple collaborators. Each collaborator has a dataset that can train a local model, multiple collaborators work together to jointly model the requested task. We used the "Three-in-One" blockchain network to archive, retrieve, and audit the joint modeling process to ensure its safety, and the consortium blockchain as the infrastructure for the distributed energy storage network. An illustration of data protection among various devices is shown in Fig. 1. We consider one TaskRequester (TR), $X$ data holders (DHs), $Y (Y \leq X)$ task collaborators selected according to SFedChain's task-related parties retrieval strategy, and $Z (Z \leq Y)$ consensus members responsible for the verification of the aggregated model. For Y collaborators, each collaborator has a local dataset $D_i = (d_1, d_2, \ldots, d_n)$, after the task requester publishes the task, the system first retrieves $Y$ collaborators related to the task from the $X$ data holders through the blockchain. The collaborators use their local data set to train to obtain the local model, and use blockchain to record the parameters of each local model. The global model (GM) is obtained using SFedChain's aggregation strategy. After continuous iterative training. Finally, the joint modeling model GM is eventually recorded in the blockchain, and the task requester obtains the result Req(GM) through the blockchain.

### SFedChain scheme

Before we introduce SFedChain, we will outline the relevant concepts and keyword definitions in SFedChain.

MasterChain: MasterChain is used to register new sites and new users, records the main configuration information of the site, manage user data access control, and store the joint modeling model. We use MasterChain to publish the requested task.

RetrievalChain: RetrievalChain is used to record the summary of site document information and the Unified Retrieval Graph that is regularly established, and it is mainly responsible for the retrieval of task-related parties.

ArgChain: The upload of local model parameters by each party and the update of the aggregated model are recorded in ArgChain in the form of transactions.

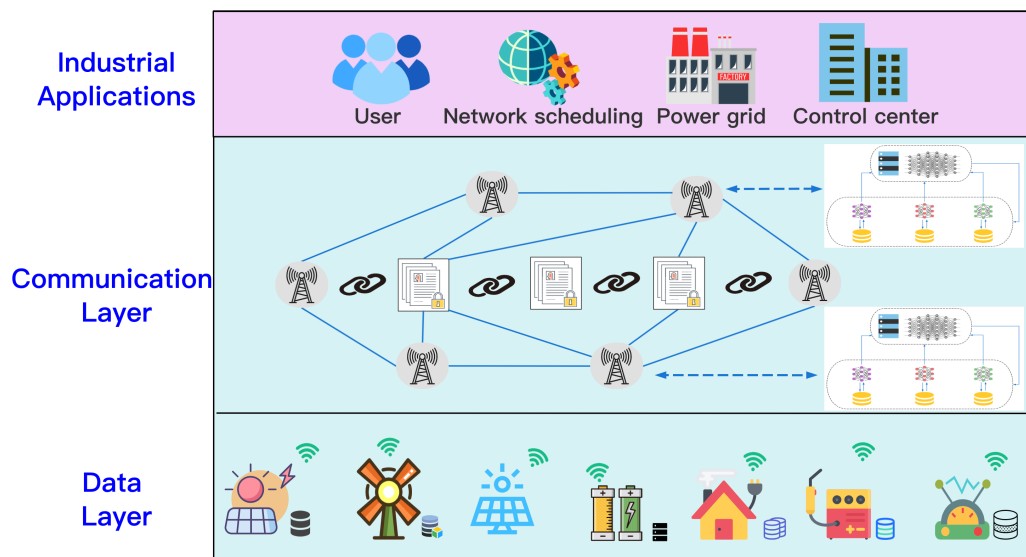

**Figure 1** **Scenario of secure multi-party data sharing.** An illustration of data protection among various devices.

DataHolder: DataHolder is a energy storage device registered with MasterChain. It has private local data and can rely on its own local dataset to train to participate in the joint modeling.

TaskRequester: TaskRequester is the user who publishes the requested task. It needs to pay the corresponding coin for the requested task to start the joint modeling.

Worker: In SFedChain, Worker is the task-related DataHolder obtained from RetrievalChain according to the requested task. It is similar to the role of participant in traditional distributed deep learning, but it can't train alone to obtain a joint modeling model, and can only use its limited dataset for local training to participate in joint modeling tasks.

Committee Members: In each round of local model parameter aggregation, $Z$ Workers with higher CreditCoin are selected from the $Y$ Workers participating in the joint modeling to form Committee Members. The Worker with the highest CreditCoin is selected as the Leader of the committee.

CreditCoin: At the beginning of the joint modeling process, SFedChain allocates the same amount of CreditCoin to the Workers participating in the joint modeling process to represent the initial credit of each Worker in the joint modeling process.

In the scenario of distributed energy storage networks, we designed the system model of SFedChain. We combined blockchain technology with traditional federated learning technology to achieve secure and efficient distributed data sharing. MasterChain was mainly responsible for issuing requested tasks and recording joint modeling models, which can improve the efficiency of the system in processing tasks. RetrievalChain was used to record the Unified Retrieval Graph that as generated regularly, and realized the quick retrieval of Workers. Workers uses its dataset to train local models, we combined the

federated learning technology and use the parameter entry method to ensure the safety of parameter sharing in the joint modeling process. Committee Members and Leader in the committee were selected for aggregation of local model parameters through SFedChain's novel aggregation strategy. Therefore, the system does not require a reliable third party for parameter aggregation, which further ensures the security of the joint modeling process. At the same time, the system introduces an incentive mechanism to encourage the active participation of DataHolder by rewarding honest participants. Eventually, TaskRequester will obtain the result of the request through MasterChain.

## Threat model

We paid attention to the secure data sharing between distributed multiple parties, and selected $Y$ Workers related to the requested task among the $X$ data providers to complete a joint modeling task. In a real-world industrial environment, task requesters and co-modeling participants are usually considered dishonest. They do not want to pay for the requested task or deliberately sabotage the joint modeling process and steal confidential information from other participants. From the above analysis, we can see that the proposed system model may face the following three threats:

1. Quality of the locally trained model: Workers may provide poorly trained model parameters due to local data set quality problems, or malicious Workers want to get rewards but do not participate in training, so they directly provide incorrect local models.

2. Instability of the parameter aggregation service: A dishonest parameter aggregation server may provide incorrect aggregation models, which will result in a serious degradation of the quality of the joint modeling model, or the parameter aggregation service may suffer malicious attacks and cause the joint modeling process to be interrupted.

3. Privacy protection in the data sharing: Workers and DataHolders participating in the joint modeling process want to obtain the parameter information of other participants through inference attacks or other methods, thereby causing user privacy leakage.

## Architecture design

The SFedChain architecture we proposed consists of three parts: MasterChain module, RetrievalChain module, and ArgChain module based on federated learning. MasterChain module establishes a secure connection between all participants, and records the joint modeling model of each requested task to achieve a rapid response to the same requested task of other users. RetrievalChain module uses the regularly generated Unified Retrieval Graph to achieve efficient retrieval of the relevant DataHolders of the requested task, and uses the retrieved Workers to implement joint modeling. ArgChain module combines with traditional federated learning to realize the secure sharing of local model parameters, and uses novel smart contract and consensus mechanism to improve the quality and efficiency of the joint modeling model. We use the ''three-chain-in-one'' architecture to achieve secure joint modeling without the original data coming out of the local situation, and maintain the system's lasting operation through all DataHolders.

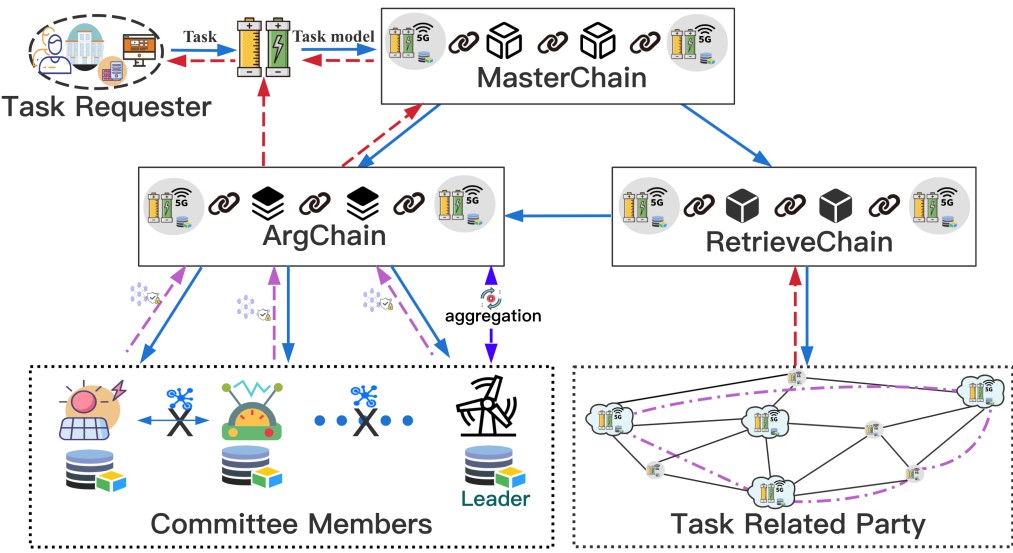

**Figure 2 Working mechanism of our proposed method.** The working mechanism of our proposed architecture.

Before a new user initiates a requested task or a new DataHolder participates in joint modeling, both should first register through MasterChain. TaskRequester publishes the requested task to MasterChain through its nearby $DataHolder_{Req}$ site server, Fig. 2 shows the working mechanism of our proposed architecture, $DataHolder_{Req}$ first searches MasterChain whether the joint modeling model of the same requested task has been recorded. If found, the system will download the joint modeling model $GM_i$ recorded in MasterChain, and then return the result of the requested task $Req(GM_i)$ to the TaskRequester. Otherwise, for a new task, the task-related information is sent to RetrievalChain to retrieve the task-related DataHolders, and then, the system will use the retrieved Workers to perform the joint modeling process through ArgChain. In each iteration, the new CreditCoin owned by each Worker is calculated based on the CreditCoin owned by each task-related parties and the local model accuracy, and then new Leader in committee and Committee Members are elected to aggregate and agree on the joint model. Finally, the result $Req(GM_{Req})$ of the requested task is returned to the TaskRequester in the form of a transaction through MasterChain. The coin paid by TaskRequester are distributed in the same proportion according to the proportion of CreditCoin ultimately owned by Workers participating in the joint modeling, in this way, more data holders will be attracted to join the system.

## CONSTRUCTION OF SFEDCHAIN SCHEME

In this section, we design and analyze the SFedChain scheme. Firstly, we design the Unified Retrieval Graph to realize the efficient retrieval of the task-related parties and protect the privacy of the DataHolder. Furthermore, we described in detail the data sharing process of our proposed model SFedChain. Finally, this article adopts the verify upload mechanism

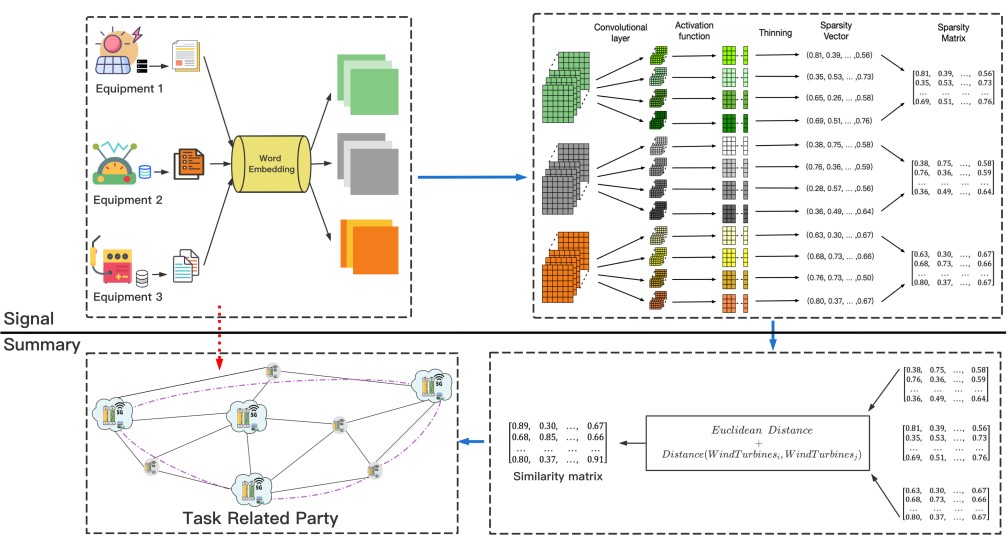

**Figure 3** **The process of building unified retrieval graph.** The legend describes the establishment of the unified retrieval graph.

of encrypted parameters and the dynamic weight consensus protocol based on CreditCoin to improve the accuracy of the joint modeling model.

## Unified retrieval graph

As a classic image data information extraction technology, CNN has been applied in many fields such as image segmentation, video classification, and target recognition (*Liu, Liu & Zhang, 2022*). For traditional CNN technology, an input image is processed by a convolutional layer, a pooling layer, and a linear layer to obtain the final result. But for text data, *Chen (2015)* proposed that the CNN model can also learn the content of text information. The output of the operating data of each device in the energy storage networks is mostly recorded in text format. How to use text data to measure the similarity of data sets between DataHolders and to achieve retrieval of task-related parties, inspired by *Liu, Guo & Chen (2021)*, we proposed the following method. The process is shown in Fig. 3.

For the processing of text data, the use of pre-trained language models (*Yamada & Shindo, 2019*; *Yamada et al., 2020*) for text characterization is considered a reliable method. We first used the selected pre-trained language model to process the text data of each DataHolder, then use the convolutional layer and the ReLU activation function to process to obtain the feature map representation of data information of each DataHolder. Finally, the sparse representation in each channel was obtained through multi-channel convolution kernel processing, $Sparsity = \frac{Number\ of\ non-zero\ values\ in\ vector}{Total\ number\ of\ vector\ elements}$, the data information owned by each DataHolder is abstracted into a matrix represented by sparsity. Since we use the sparsity expression of text statistics to retrieve Workers, it is difficult to steal the original information.

In order to continue to simplify the calculation of data similarity between DataHolders, we further processed the sparsity expression of data of each DataHolder. For the $i-th$

DataHolder, we used $DATA_i = [[d_{11}^i, d_{12}^i, ..., d_{1n}^i], ..., [d_{m1}^i, d_{m1}^i, ..., d_{mn}^i]]$ represents the data it holds, where m represents the number of texts of the filtered DataHolder. Since the Jaccard distance formula has the advantage of being independent of position and order, this article used it to calculate data similarity. Communication efficiency will also affect the creation of the Unified Retrieval Graph, the actual physical distance between DataHolders will become a factor that must be considered. Therefore, we proposed an improved Jaccard distance formula suitable for our proposed system

$$Similarity(NODE_i, NODE_j) = \frac{|NODE_i \cap NODE_j|}{|NODE_i \cup NODE_j| + \alpha * distance(NODE_i, NODE_j)} \quad (1)$$

where NODE is the sparse representation matrix of DataHolder, *distance* represents the actual physical distance between DataHolder$_i$ and DataHolder$_j$ and $\alpha$ is a hyperparameter, which is used to adjust the holding ratio of the actual physical distance.

In order to improve the efficiency of calculation and processing, we used graphs to express the relationship between DataHolders, as shown in the following definition 1:

*Definition* 1 (*Unified Retrieval Graph*): A Unified Retrieval Graph $G = \{V, E\}$ consists of a series of nodes and edges. Each vertex $V_i$ represents a DataHolder, and its weight represents the identity information of DataHolder, such as ID, data type, data size, *etc*. Each edge $E_{ij}$ connects vertices $V_i$ and $V_j$, and its weight $W_{E_{ij}}$ represents the ratio of the similarity between the two vertices to the maximum vertex similarity value ($\widehat{W_{E_{ij}}} = \frac{W_{E_{ij}}}{Max(W_{E_{ij}})}$).

Finally, with the assistance of Unified Retrieval Graph, when users submit tasks, we can find the parties involved in the task very accurately and efficiently. In order to ensure the timeliness of the unified search graph search, it is updated after performing a certain number of search operations or adding a new data holder.

## Task-related parties retrieval

The sharing of raw data will not only bring security threats, but also a series of privacy leaks. Therefore, we do not share the original data of DataHolder, and use the Unified Retrieval Graph to retrieve the task-related parties. When a TaskRequester publishes a requested task, the relevant information of the task will be recorded in the MasterChain in the form of a transaction. The MasterChain will send the processed task information to RetrievalChain to retrieve the task-related parties. The retrieval process in RetrievalChain is shown in Fig. 4.

RetrievalChain obtains $ID_{DH}$ (the ID of the DataHolder) information of nearby nodes through $ID_{TR}$ (the ID of the TaskRequester). According to the coin paid by TaskRequester, the ratio of retrieving the number of DataHolders is $ratio_{retrieval} = \frac{coin}{CONST_{coin}}$. We first traverse the Unified Retrieval Graph, and then query the vertices $V_{DH}$ representing $ID_{DH}$, retrieve its adjacent edges. If the weight $W_{E_{i,j}}$ of the adjacent edge $E_{i,j}$ is less than $ratio_{retrieval}$, the vertex $V_i$ connected by the adjacent edge $E_{i,j}$ will be included in the task-related parties set $SET_{DH}(V_j \in SET_{DH})$. After the traversal is over, the nodes contained in the set $SET_{DH}$ are the Workers participating in the joint modeling. At the same time, the equal number of CreditCoins are equally divided among Workers to identify the initial credit of each participant.

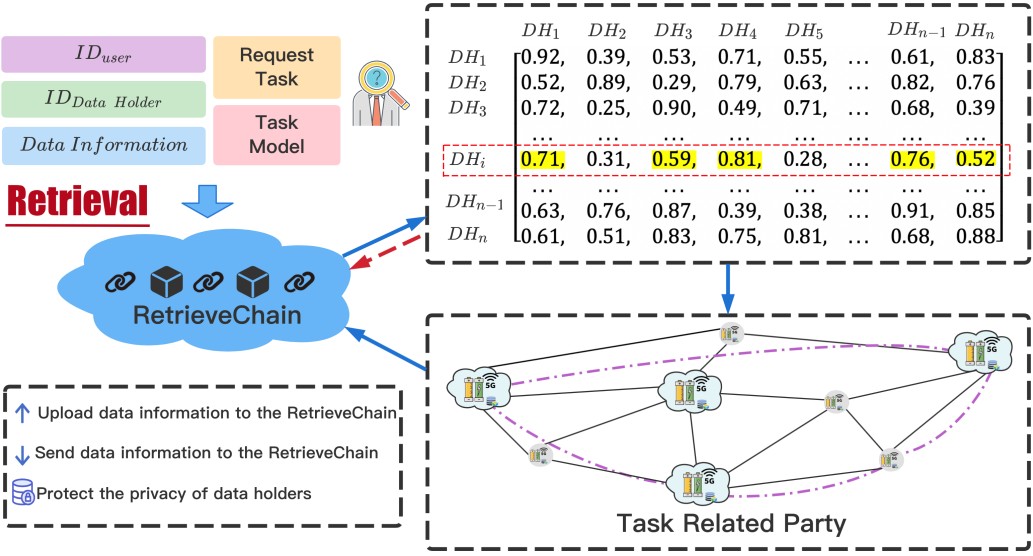

**Figure 4 Task-related parties retrieval process.** The retrieval process in RetrievalChain.

## Data sharing process

In distributed energy storage networks, the first energy storage device to join the system is responsible for the deployment of the blockchain network. Subsequent devices need to register their own nodes through MasterChain, and then use their site servers to jointly maintain the operation of the blockchain network. Before publishing the requested task, the new TaskRequester should first register as a user through the nearby device $DataHolder_{req}$, and then initialize the requested task $Req(r1, r2, \ldots, rn)$ and pay a certain amount of coins, and then $DataHolder_{req}$ submits the requested task to MasterChain in the form of a transaction. MasterChain queries its recorded historical joint model, if there is a joint model $GM_i$ for the same request task and the coin paid by TaskRequester is less than or equal to the coin spent on establishment of $GM_i$, $DataHolder_{req}$ will download the model $GM_i$ and return the result $Req(GM)$ to TaskRequester. Finally, the coin paid by the TaskRequester are distributed according to the proportion of CreditCoin owned by each Worker after $GM_i$ is jointly modeled. On the contrary, If the joint modeling model GM that matches the requested task is not found, or the GM is hit but the coin paid by the TaskRequester is more than the coin paid by the user when the $GM_i$ is established, the system will retrain the model for the requested task. During joint modeling, the system first uses the Unified Retrieval Graph to perform task-related parties retrieval through RetrievalChain, and then performs joint modeling using the retrieved Workers. After multiple iterations of training through ArgChain, the system finally obtains the joint modeling model $GM_{Req}$ that satisfies the requested task, and then uploads $GM_{Req}$ to MasterChain in the form of a transaction, and returns the result $Req(GM_{Req})$ to TaskRequester, finally the system performs the remuneration distribution of coin paid by TaskRequester.

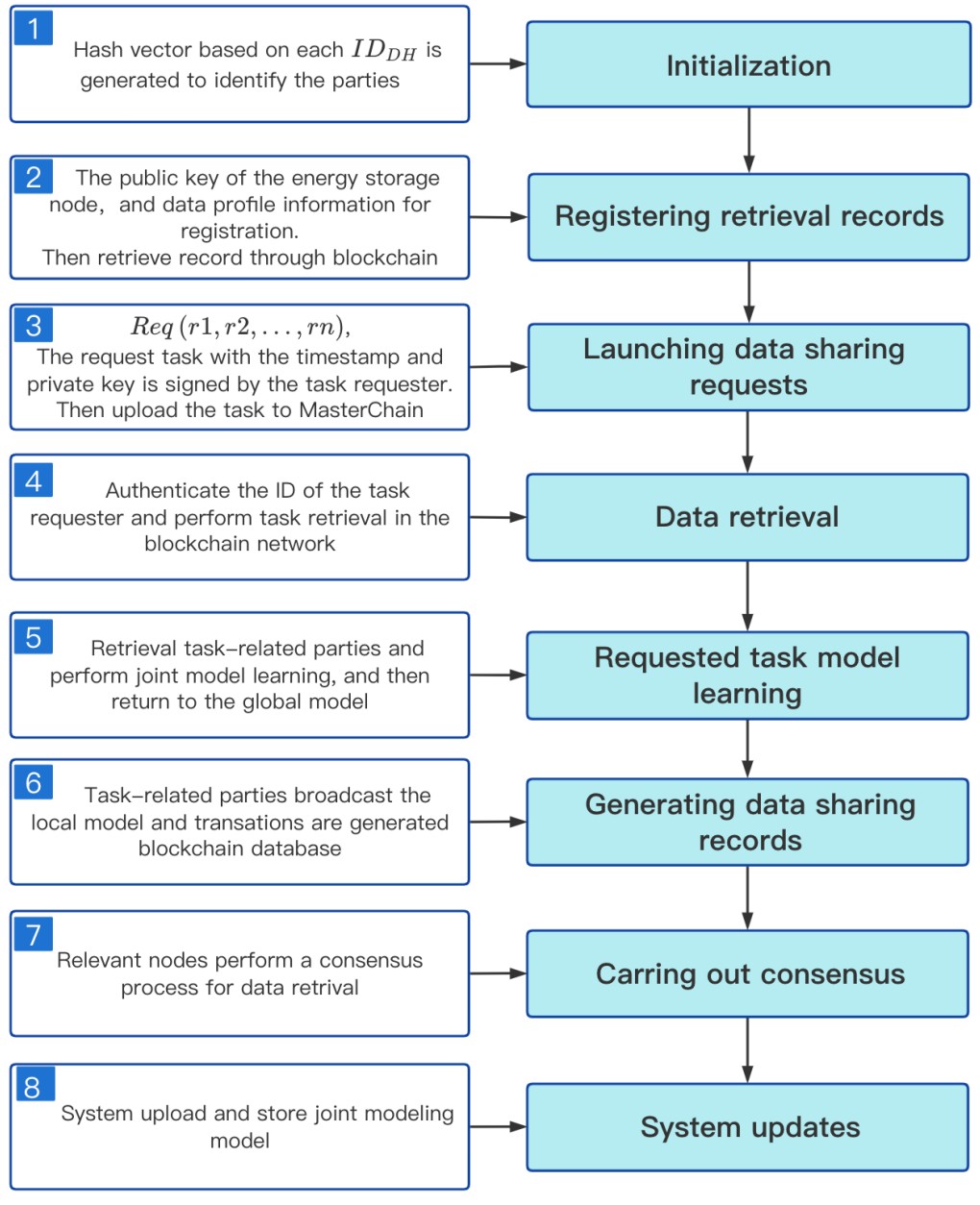

**Figure 5  Data sharing process in distributed energy storage network.**

The detailed steps of our data sharing scheme are as follows, Fig. 5 shows the process of data sharing.

1. System deployment: The first DataHolder to join the system is responsible for the deployment of the system. First, it will create MasterChain, RetrievalChain and ArgChain, and then register its node information in MasterChain. There are two main types of transactions in MasterChain: registration records of new users and new nodes, and release records of joint models. The main transaction forms in RetrievalChain

include: update records of Unified Retrieval Graph and retrieval records of task-related parties. The transaction forms in ArgChain mainly include: the upload record of local training models and aggregate model.

2. System Initialization: When a new DataHolder applies to join the system, the system distributes $ID_{DH}$ through MasterChain ($ID_{DH}$ consists of device code, module code and sensor code), and then it will work with other DataHolder to maintain the operation of the system. Similarly, a new user should first register with the nearby DataHolder before posting the requested task to obtain its unique $ID_{user}$.

3. Task Request: Task Requester submits the requested task $Req(r1, r2, \ldots, rn)$ through its nearby DataHolder $_{Req}$, and pays the corresponding coin according to the expected model effect. MasterChain checks whether the TaskRequester has been registered. If the check passes, the nearby energy storage deivce DataHolder $_{Req}$ uploads the task to MasterChain as a transaction.Otherwise, the system performs a new user registration operation.

4. Historical joint model query: Once the task submitted by TaskRequester is accepted by MasterChain, the system will query the historical union model. If there is a joint model $GM_i$ of the same requested task, the coin paid by TaskRequester is less than or equal to the coin $_i$ spent on establishment of $GM_i$, and greater than the minimum payment fee $\alpha coin_i$. Then, the result $Req_{GM_i}$ is returned to TaskRequester, and the coins paid by it are distributed according to the proportion of CreditCoin owned by each worker after $GM_i$ joint modeling. On the contrary, if the joint model $GM$ that matches the task is not found, or $GM_i$ is queried but TaskRequester paid more coins than $GM_i$ was created, retrain the joint model.

5. Retrieval task-related parties: MasterChain obtains the relevant information of the nearby energy storage device $DataHolder_{Req}$ through the requested task, and then sends the obtained $ID_{DH}$ to RetrievalChain for task related party retrieval and make the distribution of CreditCoin. Finally, RetriavalChain sends the retrieved Workers to ArgChain for joint modeling.

6. Model training: ArgChain obtains Y Workers participating in joint modeling from RetrievalChain. Each Worker uses its local dataset and initial model $GM_{Req}$ for local model training. After the local model training is over, ArgChain's smart contract algorithm will verify the local model parameters, and then upload the $W(W \leq Y)$ local model parameters that have passed the verification to the ArgChain. At the same time, the CreditCoin owned by each worker will also be adjusted according to the training quality of its local model.

7. Consensus process: We select $Z(Z = \mu Y)$ Workers with the highest accuracy from W honest workers to form Committee Members. At the same time, we select the Worker with the highest accuracy as the Leader of the committee for local model parameter aggregation, and send the aggregation results to the committee members for consensus. If the consensus is passed, the Leader of the committee will release the aggregation model $GM_{Req}$ and upload it to AgrChain, which can facilitate the Workers participating in the joint modeling process to download and update their local models.

8. Complete the requested task:After several iterations of training, the joint modeling model $GM_{Req}^{final}$ is finally established. According to the proportion of CreditCoin held by each Worker, the system distributes the coin paid by TaskRequester as reward to Workers participating in joint modeling, which can encourage DataHolder to actively participate in joint modeling of requested task next time. Finally, the system will upload and store the joint modeling model $GM_{Req}^{final}$ to MasterChain, and return the result $Req(GM)$ to the TaskRequester.

## SFecChain aggregation strategy

For a DataHolder, it is difficult to guarantee the training quality of the local model due to its limited resources, and to effectively protect the privacy of users by sharing the original data information of all parties for centralized training. As a result, the aggregation strategy of SFecChain not only expands the amount of data required but also protects data privacy of users by integrating multiparty DataHolders' local model parameters for joint model training without the original data being local.

In the local model parameter aggregation stage, dishonest Workers may upload incorrect local model parameters, and it is difficult to guarantee the reliability of the Leader in committee responsible for local model parameter aggregation. Therefore, it is difficult to establish an accurate and efficient joint model. Existing consensus protocols, such as PoW, etc. miner requires huge computational overhead to solve cumbersome data problems, and its long consensus process seriously affects the efficiency of system modeling, so it is not suitable for scenarios with frequent transactions. To solve these problems, inspired by *Tang, Zhang & Hu (2020)*, we proposed a credit-based dynamic weight consensus protocol combined with deep learning. The system can effectively identify dishonest Workers participating in each round of joint modeling, and can also use the historical credit of the Workers participating in joint modeling and the accuracy of each round of local model training, and combine with the way of dynamically selecting committees for consensus. Eventually a high-quality joint model will be obtained.

### Encrypted parameter upload

In order to ensure the secure sharing of local model parameters during the joint modeling, we integrate the differential privacy mechanism into SFedChain. The privacy of Workers is protected by adding noise to local model parameters. We use smart contract of ArgChain to filter the malicious behavior of dishonest participants in the joint modeling, which effectively guarantees the quality of the joint training model, and combined with the distributed ledger technology of ArgChain to further ensure the security of local model parameter sharing.

After the local training of the Workers participating in the joint model training is completed, we add noise that conforms to the Laplace distribution to it, and then upload it to ArgChain. The privacy of users is protected through a differential privacy mechanism. We use the selected random algorithm $\mathcal{L}$. For any two local model parameters $LM_i$ and $LM_j$ participating in joint modeling, we make them satisfy:

$$\Pr[M(LM_i) \in S] \le e^{\varepsilon} \Pr[M(LM_j) \in S] + \delta$$

The system provides differential privacy protection for the local model parameters $LM_i$ and $LM_j$ satisfying $\varepsilon$. Specifically, we add random noise with a density function of $f(x|u,b) = \frac{1}{2b}e^{-\frac{x-u}{b}}$ that obeys the Laplace distribution to the local model parameters $LM_i$ and $LM_j$, and select a suitable differential privacy budget $\varepsilon$ to obtain the processed local model parameters $\widehat{LM_i}$ and $\widehat{LM_j}$. A good trade-off between model quality and privacy protection is achieved.

---

**Algorithm 1** Differential private parameter verification

---

**Input:** Workers participating in joint modeling $P$

**Output:** Workers who have passed ArgChain smart contract verification $\hat{P}$

1: $avg = \frac{\sum_{i=1}^{length(P)} P_i.accuracy}{length(P)}$

2: $threshold = avg - u \times (max(P.accuracy) - avg)$

3: **while** each Worker $p_i \in P$ **do**

4:     **if** $p_i.accuracy \geq threshold$ **then**

5:         $\hat{P} \leftarrow p_i$

6:     **end if**

7:     Add random noise that obeys Laplace distribution to construct a local model that satisfies $\varepsilon$ differential privacy protection $\hat{p}_i.model$

8: **end while**

9: **return** $\hat{P}$

---

Since Workers participating in joint modeling may upload local model parameters of poor quality, or dishonest Workers maliciously upload incorrect local model parameters, these behaviors will seriously affect the quality of the final joint training model. Therefore, it is particularly important to screen non-compliant local model parameters. In response to the above problems, we propose a differential privacy protection mechanism for local model parameters and a parameter verification mechanism based on ArgChain's smart contracts. Algorithm 1 illustrates the verification process of the smart contract algorithm of ArgChain.

### Dynamic weight consensus protocol based on credit

After the local model parameters of Y Workers participating in the joint modeling are verified by the smart contract of ArgChain. Package the verified local model parameters $\hat{Y} (\hat{Y} \leq Y)$, and then upload them to ArgChain in the form of transactions. The process for training work based consensus is illustrated in Fig. 6. The local model parameters in $Y$ in each round will be aggregated by the miners. A conventional idea is that the Worker with the highest accuracy of local model training in each round has high credibility and can be used to aggregate local model parameters. This is considered a greedy strategy. The training accuracy of local model is not necessarily related to whether it is honest or not, and the reliability of the aggregation process cannot be guaranteed. In order to ensure the accuracy of local model parameter aggregation, we propose a credit-based dynamic weight consensus protocol.

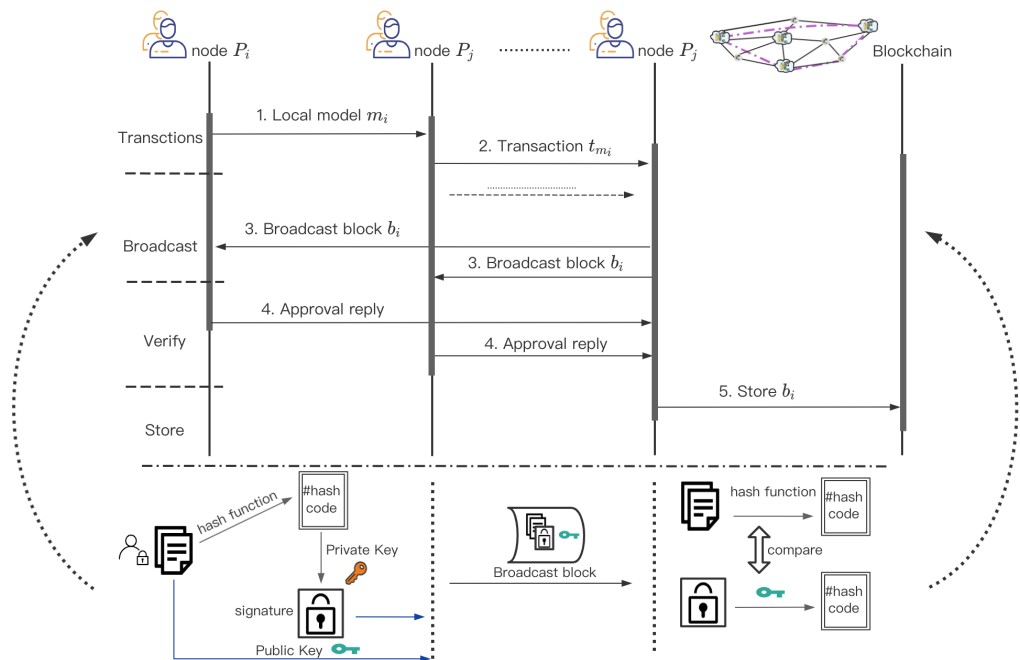

**Figure 6 Consensus process of dynamic weight consesus protocol.** The process for training work based consensus.

We comprehensively consider the historical credit of the Y Workers participating in the joint modeling and the training accuracy of each round of local model, and then we calculate the new credit value of the Y Workers in the new round. The Leader in committee and Z Committee Members are elected through the new credit value for the aggregation of local model parameters and the consensus of the aggregation results. Algorithm 2 illustrates the overall process of credit-based dynamic weight parameter aggregation.

# SECURITY ANALYSIS AND PERFORMANCE EVALUATION

We have established a multi-party data security sharing and privacy protection mechanism in distributed energy storage networks by applying blockchain technology. By integrating into the traditional federated learning mechanism, the "Three Chains in One" structure has been established. We solved the threat model proposed in the "Threat model" section.

1. Security proof for SFedChain: The traditional single chain structure of blockchain is difficult to meet the requirements of data retrieval, calculation and privacy protection at the same time. Therefore, we propose a "three chains in one" architecture including MasterChain, RetrievalChain, and ArgChain. MasterChain is mainly responsible for the publication of events of requested task and the quick query of historical aggregation models. RetrievalChain is mainly responsible for regular update of the Unified Retrieval Graph. ArgChain mainly carries out the secure sharing of parameters of the federated learning. They perform their respective duties to improve the network performance of the system to achieve data security sharing and privacy protection.

---

**Algorithm 2** Credit-based dynamic weight parameter aggregation

---

**Input:** Workers participating in joint modeling $P$, Workers verified by ArgChain $\hat{P}$, iteration times $iter = 1$

**Output:** Encrypted global model, $GM$

1: **if** $length(verified(P)) = length(P)$ **then**
2:     **while** each participant $p_i \in P - \hat{P}$ **do**
3:         $p_i.CreditCoin = p_i.CreditCoin - u \times p_i.CreditCoin$
4:         $sum_r = sum_r + u \times p_i.CreditCoin$
5:     **end while**
6:     **while** each participant $p_i \in \hat{P}$ **do**
7:         $p_i.CreditCoin = p_i.CreditCoin + \frac{sum_r}{length(\hat{P})}$
8:     **end while**
9:     **while** each participant $p_i \in \hat{P}$ **do**
10:         $S_i^{CreditCoin} = \frac{e^{p_i.CreditCoin}}{\sum_{j=1}^{length(\hat{P})} e^{p_j.CreditCoin}}$
11:         $S_i^{Acc} = \frac{e^{p_i.accuracy}}{\sum_{j=1}^{length(\hat{P})} e^{p_j.accuracy}}$
12:     **end while**
13:     $C = S_{CreditCoin} + S_{Acc}$
14:     **while** each $c_i \in C$ **do**
15:         $S_i^C = \frac{c_i}{\sum_{j=1}^{length(C)} e^{c_j}}$
16:     **end while**
17:     **while** $iter \leq length(\hat{P})$ **do**
18:         Get the $P_k$ corresponding to the maximum value in $S_C$ as Leader in committee, and get the $\widetilde{P}$ corresponding to the larger $\mu \times length(\hat{P})$ values in $S_C$ to form Committee Members for consensus
19:         $GM = \sum_{j=1}^{length(\hat{P})} S_j^C \times P_j.weight$
20:         **if** $consensus(GM) = true$ **then**
21:             **return** $GM$ to ArgChain;
22:         **else**
23:             $\hat{P} = \hat{P} - P_k$
24:         **end if**
25:         $iter = iter + 1$
26:     **end while**
27:     **return** 0
28: **end if**

---

2. Smart contract verification mechanism of ArgChain combined with differential privacy: Before uploading the trained local model parameters to ArgChain, the real data can be hidden by perturbing the local model parameters and adding noise that conforms to the Laplace distribution. Attacks with background knowledge can be avoided to obtain the original data information of the DataHolder. The trained local model parameters are verified through smart contract of ArgChain. Since DataHolder may train poor

quality model parameters or dishonest DataHolder maliciously upload incorrect local model parameters, these factors will lead to lower quality of the joint model. Filtering the uploaded local model parameters through ArgChain's smart contract can guarantee the training quality of the joint model.

3.  No fixed aggregation server: In the aggregation phase of the local model parameters. A dishonest parameter aggregation server generates incorrect aggregation parameters, or the parameter aggregation server is attacked by a malicious attacker, which may cause the interruption of the parameter aggregation process. We propose a method of dynamically selecting parameter aggregation server and Committee Members based on the credit of Workers to perform parameter aggregation services, and to agree on the result of parameter aggregation to ensure the safety and accuracy of the parameter aggregation process.

4.  The quality of the joint training model: In order to obtain a higher-quality joint training model, we propose a local model parameter aggregation algorithm based on dynamic weight allocation. Specifically, the new CreditCoin owned by each Worker is calculated by using the CreditCoin owned by the Worker and the accuracy of each round of local training. Perform softmax on the new CreditCoin, and perform a weighted summation of its local model parameters according to its different proportions to obtain the joint model parameters for each round.

5.  Incentive mechanism: In order to ensure the durable operation of SFedChain and attract more DataHolders to participate in the joint modeling of the requested task. We propose to pay rewards for participating in joint modeling workers to attract more DataHolders to join. TaskRequester pays Coin for its requested task. According to the performance of Workers' joint modeling process, different amounts of rewards are allocated to improve the enthusiasm of DataHolder to participate.

## Evaluation setup

We used 20 Newsgroups and AG News to simulate the adaptability and high efficiency of SFedChain, which are international standard datasets that are often used to evaluate text-related machine learning algorithms. The 20 newsgroup dataset is a collection newsgroup documents. This dataset collected about 20,000 newsgroup documents, which were evenly divided into 20 newsgroup collections with different topics. It has become a popular data set for experiments in text applications of machine learning techniques. The AG news is a collection of more than one million news articles. News articles were gathered from more than 2,000 news sources by ComeToMyHead in more than one year of activity. The dataset includes 120,000 training samples and 7,600 test samples. Each sample was a short text with four types of labels. We used these two datasets to simulate the text data generated by the equipment operation and monitoring of each device.

We simulated different numbers of energy storage devices, which have their own local dataset and can be independently modeled. We used the selected dataset to split the data entries, and regroup according to the number of groups set in each experiment, to simulate DataHolders in SFedChain. We used text topic classification analysis to simulate the requested task of the TaskRequester, and implemented our improved attention mechanism

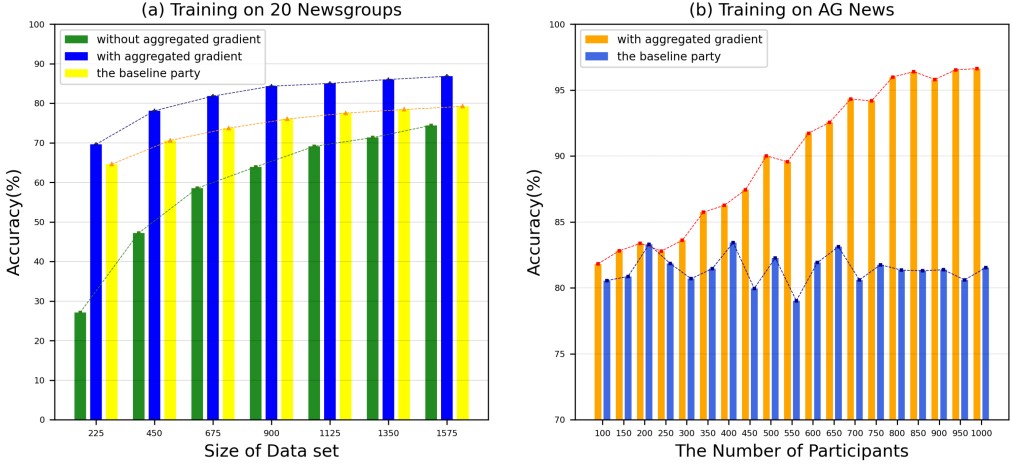

**Figure 7** **(A–B) Accuracy in various datasets.** (A) Training on 20 Newsgroups dataset. (B) Training on AG News dataset. The performance comparison between our proposed model and the benchmark method text graph convolutional networks on various datasets.

on text data to perform the joint modeling process of the SFedChain scheme in the process of distributed multi-party data sharing.

We perform a lot of simulations on Linux ubuntu 4.15.0-45-generic, and the hardware conguration is as follows: Intel(R) Xeon(R) Gold 5218 CPU @ 2.30 GHz, 251G, 10TB hard drive, interpreter Python 3.8.10, and pytorch 1.7.0, We analyzed and evaluated the performance of the SFedChain scheme, and gave the following experimental results.

## Numerical results

We use 20 Newsgroup and AG News benchmark datasets to evaluate the accuracy of our proposed model. In order to ensure the accuracy of the experimental results, we conducted 10 experiments and took the average of the results. The performance comparison between our proposed model and the benchmark method text graph convolutional networks (*Yao, Mao & Luo, 2019*) on various datasets is shown in Fig. 7. From Fig. 7A, we can see that compared to the benchmark method, most of our test groups have obtained a higher accuracy, which shows that our proposed SFedChain has a high diagnostic ability. At the same time, we can see that the more data that each DataHolder has, the higher the accuracy of the joint model built together. This is because that the accuracy of the model has a certain relationship with the number of datasets and computing resources owned by the DataHolder in the actual environment. Figure 7B shows the accuracy results with various number of data providers. As data providers increase, the accuracy curve of the model first grows and eventually stabilizes. This means that the lack of data volume affects the accuracy of the model to a certain extent. Eventually, as the data volume saturates, the model reaches the limit of its diagnostic ability. Therefore, we determine the number of Workers according to the minimum and maximum payment amount of the request task to adapt to the model's diagnostic ability.

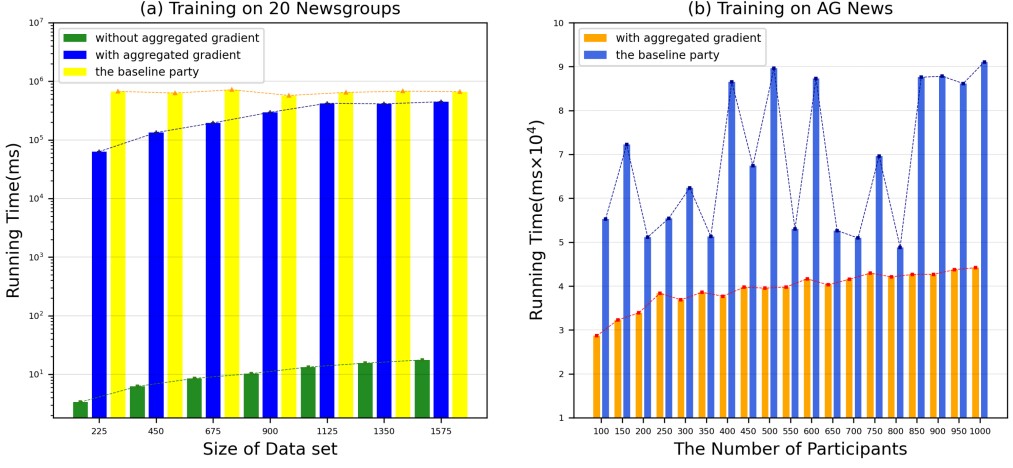

**Figure 8 Running time in various datasets.** The running time of our proposed mechanism in different subdatasets.

However, the time to build a model is also a key factor to measure the performance of the model. Therefore, we evaluated the time for each DataHolder to build the local model and the joint model, and compared it with the benchmark method text graph convolutional networks model. Figure 8A shows that the running time of our proposed mechanism in different subdatasets. The results show that compared with the benchmark method, the time spent on establishing the joint model of our proposed scheme is significantly reduced, and it can well meet the waiting time of the TaskRequester. From Fig. 8B, we can see that as the DataHolder increases, its running time can still remain stable. The processing time spent to establish the joint model will not change significantly due to the continuous addition of DataHolder, which shows that the model we proposed has good compatibility. Due to the existence of malicious workers, the performance of the joint model is affected to different degrees. Therefore, we simulate the anti-interference of our model by simulating different proportions of malicious attackers to conduct simulation experiments. We use the AG News dataset as the experimental dataset to simulate a scenario of joint modeling of 50 energy storage nodes, and simulate malicious nodes by modifying the dataset in the nodes to be mismatched label pairs. We set different proportions of malicious attackers: attack strength 10%, attack strength 20%, attack strength 30%, attack strength 40%, attack strength 50%. From Fig. 9, we can see that the presence of a small number of dishonest nodes does not affect the accuracy of our proposed model for joint modeling. The system can dynamically distinguish malicious nodes through the dynamic weight consensus protocol and smart contract mechanism to ensure the quality of the training data set, thereby effectively improving the performance of the system.

Through the above evaluation, we can observe that with the addition of the new data holder, the accuracy of the joint modeling model can be continuously improved without significantly increasing the time spent in the joint modeling process. So our scheme can attract more data holders to join to improve the joint modeling effect of the request task.

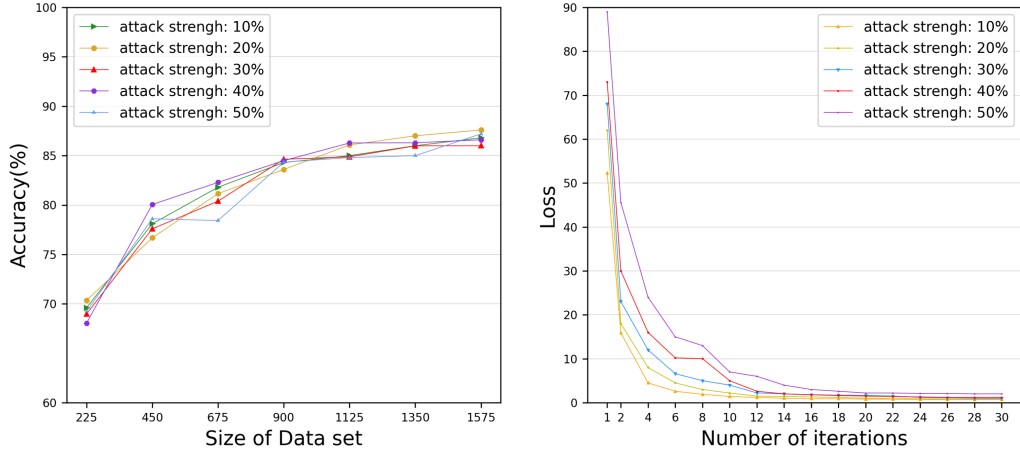

**Figure 9 Accuracy and Loss under various attack strengths.** The anti-interference of our model by simulating different proportions of malicious attackers.

As the number of data holders participating in joint modeling increases, the system needs to perform more local model aggregation and updates, which causes a slight increase in system overhead. However, with the addition of more data holders, the data scale of joint modeling is further improved, and SFedChain's secure data sharing mechanism brings a significant increase in the performance of joint modeling models, which effectively improves the quality of service in distributed energy storage networks.

## CONCLUSION

In this article, we proposed a blockchain-based machine learning scheme for privacy data sharing in distributed energy storage networks. A series of security analysis and simulation experiments show that our proposed scheme not only protects data privacy, but also further improves the accuracy of the joint modeling model in energy storage device applications through a secure data sharing mechanism. The combination of blockchain and machine learning is an effective way to realize the safe sharing of data. However, the question of how to use blockchain technology to further ensure privacy protection in the data sharing process is still worthy of attention, as is determining how to gather more valuable data information from distributed multi-party data holders. Therefore, machine learning algorithms suitable for joint modeling scenarios still need further research. In addition, due to the limitation of communication bandwidth, determining how to further reduce the communication overhead of the joint model modeling process remains to be further discussed.

## ADDITIONAL INFORMATION AND DECLARATION

### Funding

This work was supported by the State Grid Corporation Headquarters Science and Technology Project ''Research on Key Technologies to Support Network Operation of Distributed Energy Storage'' under grant 5100-202199544A-0-5-ZN. The funders had no role in study design, data collection and analysis, decision to publish, or preparation of the manuscript.

### Grant Disclosures

The following grant information was disclosed by the authors:
The State Grid Corporation Headquarters Science and Technology Project ''Research on Key Technologies to Support Network Operation of Distributed Energy Storage'': 5100-202199544A-0-5-ZN.

### Competing Interests

The authors declare there are no competing interests.

### Author Contributions

- Mingming Meng conceived and designed the experiments, performed the experiments, analyzed the data, performed the computation work, prepared figures and/or tables, authored or reviewed drafts of the article, and approved the final draft.
- Yuancheng Li conceived and designed the experiments, performed the experiments, analyzed the data, performed the computation work, prepared figures and/or tables, authored or reviewed drafts of the article, and approved the final draft.

### Data Availability

The SFedChain: blockchain-based federated learning scheme is available in GitHub: https://github.com/mengmingming-sudo/SFedChain.git.

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
