# Peer review of "SFedChain: blockchain-based federated learning scheme for secure data sharing in distributed energy storage networks"

_PeerJ Computer Science, doi:10.7717/peerj-cs.1027_

## Round 0.1 · original submission · Minor Revisions

Based on the reports of reviewers, both of them believed that the paper is well-written with a solid contribution. Some writing issues should be corrected before accepting it as the English grammar issues are pervasive. A minor revision is needed to enhance the quality.

Reviewer 1 ·

Basic reporting

This paper provides a comprehensive review of research work on privacy-preserving data sharing in the introduction and related literature, and introduces the importance of secure joint modeling in distributed energy storage networks. This paper innovatively proposes a scheme called SFedChain based on blockchain and federated learning to realize the safe sharing of data in distributed energy storage network, which has strong practicability and logic. This paper is innovative in application, considers the scenario of data sharing in distributed energy storage networks, and expounds the importance of data privacy protection. In terms of method and theoretical innovation, this paper realizes efficient joint modeling by combining federated learning to transform data sharing problems into machine learning problems, and uses blockchain technology to record to ensure security, making it more practical. In joint modeling, it is logical to ensure the security and improve the accuracy of joint modeling through the designed encryption parameter upload review mechanism and reputation-based dynamic weight consensus protocol. The experimental results demonstrate the effectiveness and applicability of the method. The research problem in this paper is meaningful, and the method is innovative. The sequence structure is clear and the paper has a good organization structure. However, the manuscript still needs to be modified before it is published. I have several questions listed as follows:
1)There are some Language problems and typos or grammar errors in this manuscript. Many of them could cause difficulties to the readers.
2)The author proposes to use the consortium chain to trace the source of the joint modeling process. Why does this article apply the consortium chain instead of the public chain or private chain? Please give the author a detailed explanation.
3)In the simulation of the attack experiment in Figure 9, the authors should give a more detailed experimental setup.

Experimental design

In order to solve the problem of effective data management in the distributed energy storage network, this paper establishes a multi-party data security sharing and privacy protection mechanism, and effectively solves the possible threats to the model through a series of security analysis. This paper uses real benchmark data sets and conducts comparative test analysis to verify that the improved blockchain consensus mechanism and incentive algorithm can effectively improve the accuracy of federated learning. By simulating different proportions of attack intensities, the anti-interference ability of the model is verified. The method has certain novelty.

Validity of the findings

In the experimental part of this paper, the theoretical results are reasonable. For the proposed threat model, a reliable solution is provided through security analysis. At the same time, benchmark experiments and comparative verifications demonstrate the effectiveness and robustness of the model, and realize the secure sharing and privacy protection of data in distributed energy storage networks.

Reviewer 2 ·

Basic reporting

In this work, the authors have proposed a deep learning scheme called SFedChain in distributed energy storage networks to ensure the secure sharing of data between distributed devices and improve the training accuracy of federated learning models.
In the new scheme, the three-in-one blockchain architecture is designed for on-chain storage and computation of a large number of model parameters, which is different from the current small-capacity data storage method mainly based on ledgers. It provides an effective solution to the big data storage bottleneck problem of model training parameters in energy storage scenarios.
In addition, this scheme designs a secure federated learning model to ensure the security of power data sharing among distributed energy storage nodes. It also solves the problem of aggregator single-point failure and malicious attack in traditional federated learning, and effectively improves the accuracy of the joint model through the designed dynamic weight method.
Finally, a series of safety analysis proved the reliability of the scheme. Comparative experiments and performance analysis show the efficiency and anti-interference of the scheme.
This paper is well–written. The proposed scheme is quite rational and has a rigorous theoretical reasoning process. The originality is clear. The technology adopted is advanced. The objective evaluation results are reported. The solution is suitable for problem solving in distributed energy storage networks. Hence I would like to recommend to accept the paper for publication.

Experimental design

The experimental design is good.

Validity of the findings

The findings are valid.

Additional comments

1, In the first paragraph of chapter 1, the “blockchain” is written as “blockchian”. Please proofread carefully.

2, In section 2 of chapter 2, the explanation of 〖ID〗_DH and 〖ID〗_TR are not given when it appears at the first time.

3, In chapter INTRODUCTION, page 2, line 99 - "privacy", use capital "P". Line 118, "In Section 2, We give implementation", Use the small letter "we".

4, The text in Figure 1 is too small, please use a high-quality vector image and adjust the font size.

---

## Round 0.2 · accepted · Accept

The reviewer did not have further comments for the revised version. Therefore I recommend accepting this manuscript.

Reviewer 2 ·

Basic reporting

I have no comment.

Experimental design

I have no comment.

Validity of the findings

I have no comment.